# Soil Properties Regulate Soil Microbial Communities During Forest Succession in a Karst Region of Southwest China

**DOI:** 10.3390/microorganisms12112136

**Published:** 2024-10-24

**Authors:** Shanghua Jiang, Min Song, Hu Du, Feng Wang, Tongqing Song, Huijun Chen, Fuping Zeng, Wanxia Peng

**Affiliations:** 1Institutional Center for Shared Technologies and Facilities of Institute of Subtropical Agriculture, Chinese Academy of Sciences, Changsha 410125, China; 18229595279@163.com (S.J.); songmin_summer@163.com (M.S.); hudu@isa.ac.cn (H.D.); wth19810724@163.com (F.W.); songtongq@isa.ac.cn (T.S.); 18772285461@163.com (H.C.); fpzeng@isa.ac.cn (F.Z.); 2Huanjiang Agriculture Ecosystem Observation and Research Station of Guangxi, Guangxi Key Laboratory of Karst Ecological Processes and Services, Huanjiang Observation and Research Station for Karst Ecosystems, Chinese Academy of Sciences, Huanjiang 547100, China; 3College of Environment & Ecology, Hunan Agricultural University, Changsha 410128, China; 4Institute of Agricultural Environment and Ecology, Hunan Academy of Agricultural Sciences, Changsha 410125, China

**Keywords:** soil microbial community, diversity, occurrence, succession, karst forest

## Abstract

Natural vegetation restoration has emerged as an effective and rapid approach for ecological restoration in fragile areas. However, the response of soil microorganisms to natural succession remains unclear. To address this, we utilized high-throughput sequencing methods to assess the dynamics of soil bacterial and fungal communities during forest succession (shrubland, secondary forest, and primary forest) in a karst region of Southwest China. Our study revealed that bacterial α-diversity was significantly higher in secondary forest compared to both shrubland and primary forest. Intriguingly, the soil bacterial community in primary forest exhibited a closer resemblance to that in shrubland yet diverged from the community in secondary forest. Conversely, the soil fungal community underwent notable variations across the different forest stages. Furthermore, analysis of the microbial co-occurrence network revealed that, within these karst forests, the relationships among soil fungi were characterized by fewer but stronger interactions compared to those among bacteria. Additionally, soil properties (including pH, soil organic carbon, total nitrogen, moisture, and available potassium), soil microbial biomass (specifically phosphorus and nitrogen), and plant diversity were the drivers of soil bacterial community dynamics. Notably, soil pH accounted for the majority of the variations observed in the soil fungal community during karst forest succession. Our findings provide valuable insights that can inform the formulation of strategies for ecological restoration and biodiversity conservation in karst regions, particularly from a microbial perspective.

## 1. Introduction

Belowground biodiversity, notably microbial diversity, plays a crucial role in regulating aboveground biodiversity and ecosystem functioning [1]. Furthermore, microbial diversity is a key driver of multifunctionality in terrestrial ecosystems, encompassing critical aspects such as climate regulation, soil fertility, material production, and provision [2,3]. These contributions ultimately enhance human well-being [4]. As a vital component of terrestrial ecosystems, microbial biodiversity in forests has garnered attention comparable to that of macro-organisms such as plants [5,6]. Furthermore, it has been suggested to apply existing macro-ecological theories to the field of soil microbial ecology, as proposed in [7]. Various environmental factors play regulatory roles in shaping soil microbial diversity. For instance, temperature and soil carbon content influence soil archaea, while aridity, vegetation characteristics, and soil pH regulate bacterial communities, as reported in [8]. During secondary succession, the soil bacterial community is affected by multiple factors, including plant diversity and composition, as well as soil nutrients such as total organic carbon and total nitrogen, as detailed in [9]. Additionally, soil fungal diversity and functionality are influenced by plant species during afforestation processes [10]. When compared to arable land, forest ecosystems exhibit more stable and complex microbial networks [11].

The karst landscape in the southwest of China spans over 0.54 million km^2^, constituting one of the three largest continuous distribution areas in the world [12]. This landscape is particularly susceptible to disturbances, exhibiting unstable characteristics, and posing challenges for self-adjustment [13,14,15], partly due to slow species turnover and infertile soils [16]. Despite these challenges, substantial increases in vegetation growth and carbon stocks have positioned the karst region as a hot spot of global greening [17], with arbuscular mycorrhizal fungi playing a potentially crucial role in the maintenance of multiple ecosystem functions [18]. Recent research [7] has demonstrated that microbial communities in forest ecosystems undergo distinct changes across successional stages. Specifically, early successional stages tend to be dominated by bacteria (r-strategists), whereas late successional stages are prone to be dominated by fungi (K-strategists). This shift may be attributed to the differing responses of bacteria and fungi to succession [19], influenced by factors such as size, growth, and turnover rates [20]. Additionally, some studies [21] have indicated that karst forests exhibit greater connectivity among bacterial and fungal communities compared to non-karst forests, suggesting that increased microbial diversity strengthen the complexity of co-occurrence networks. Despite the growing interest in understanding the dynamics of soil microbial communities during forest succession, our knowledge of their response magnitude and direction in karst forest succession, characterized by specific plant communities and soil properties [22], remains limited. In particular, the specific drivers of microbial community changes in different succession stages in karst regions in southwest China are still poorly understood. Gaining these insights into their drivers is vital to increase ecosystem stability and function, particularly in the context of achieving international carbon sequestration and carbon neutrality goals.

To bridge this knowledge gap [9,18,19,21], we conducted a study in the karst region of Southwest China. In this study, we employed the sequencing of 16S rRNA gene and ITS gene amplicons to gain insights into the community composition and diversity of soil bacteria and fungi across various stages of forest restoration, including shrubland, secondary forest, and primary forest. Subsequently, we quantified the soil microbial community, its diversity, and the co-occurrence network in relation to karst forest succession. Our ultimate goal was to identify the key factors influencing soil microbial dynamics during the process. This research contributes to a deeper understanding of the critical role that soil microorganisms play in maintaining ecosystem functions and services in a typical karst environment.

## 2. Materials and Methods

### 2.1. Study Area

The study area was located in Huanjiang Maonan Autonomous County, Guangxi (107°51′–108°43′ E, 24°44′–25°33′ N), characterized by a subtropical monsoon climate with an annual average temperature of 19.3 °C, 1451.1 h of annual average sunshine hours, and 1529 mm of annual average precipitation. In the karst region, shrubland, secondary forest, and primary forest represent three typical stages of forest succession stages. The respective plots for these forests were established at Mulian Karst Experimental Station (Mulian), Guzhou in Xianan Township (Guzhou), and the Mulun National Nature Reserve (Mulun) (Figure A1). In the shrubland plot in Mulian, the dominant woody vegetation includes *Vitex negundo* L., *Alangium chinense* (Lour.) Harms, and *Ligustrum quihoui* Carrière [22], which have naturally regenerated since abandonment in 1985. In the secondary forest plot in Guzhou, with less disturbance due to the application of projects of comprehensive control for rocky desertification, the dominant species are *Bauhinia brachycarpa* Thunb., *Cipadessa baccifera* Miq., *Radermachera sinica* (Hance) Hemsl., and *Toona sinensis* (A.Juss.) M.Roem. [22]. Meanwhile, the primary forest plot at Mulun is dominated by *Cryptocarya concinna* Hance, *Itoa orientalis* Hemsl., and *Brassaiopsis glomerata* (Blume) Regel [22]. Mulun National Nature Reserve boasts the best preservation and the largest primary karst forest, which is a mixed evergreen and deciduous broadleaf forest [16]. The soil type in all the three regions is Leptosols, and the site conditions such as slope and slope aspect are consistent across the plots.

### 2.2. Vegetation Investigation

In 2007, dynamic forest plots of shrubland, secondary forest, and primary forest, each with a size of 220 m × 40 m, extending from the valley to the hilltops, were established in Mulian, Guzhou, and Mulun, respectively. These plots were subsequently divided into 22 quadrants of 20 m × 20 m, and each quadrant was further divided into 16 sub-quadrants with a size of 5 m × 5 m, following the standard protocol set by the Center for Tropical Forest Science (https://forestgeo.si.edu (accessed on 1 January 2024)). All woody plants with a diameter at breast height (DBH) exceeding 1 cm were tagged, identified, measured, and georeferenced. Inventory was conducted every five years.

For our analysis, we utilized woody plant inventory data from 2017, specifically focusing on the central area of each 20 m × 20 m quadrant across the three plots. We used the average DBH, species richness, and Shannon-wiener index as vegetation factors. The average DBH was calculated based on all woody plants within each plot. The determination of the richness index and Shannon index were conducted according to the reference [22].

### 2.3. Soil Sample Collection and Determination

In October 2019, we collected soil samples every 20 m (i.e., the middle sample point) along the middle sample line of the plot from bottom to top, resulting in a total of 33 samples. The soil temperature and volumetric water content were measured 5–8 times around each sampling point using a soil time domain reflectometer (TDR200, Beijing Intell-sun Technology Co., Ltd., Beijing, China), and average values were calculated to represent these measurements. Eight to ten surface soil samples (0–15 cm) around the sampling points were combined after thoroughly mixing to form a composite soil sample. About 150 g of soil from each point was kept in a liquid nitrogen tank and transported to the laboratory for the high-throughput sequencing of soil microorganisms. We screened soil samples (about 500 g) to remove roots and stones through a 10-mesh sieve. One part was stored in a freezer (4 °C) for the measurement of soil ammonium nitrate and microbial biomass C, N, and P, while the other part was prepared for the determination of soil physio-chemical properties.

Soil pH was determined by a pH meter using a 1:5 soil/water suspension. The determination of soil organic carbon (SOC), total nitrogen (Total N), and total phosphorus (Total P) adopted the methods of Walkley–Black wet oxidation, semi-micro Kjeldahl, alkali digestion–molybdenum antimony colorimetry, and flame photometry, respectively [23]. The determination of available nitrogen (AN), available phosphorus (AP), and available potassium (AK) adopted the alkaline hydrolysis diffusion method, Olsen method, and OAc method, respectively [23]. The exchangeable Ca^2+^ and Mg^2+^ were determined by the EDTA titration method, and nitrate nitrogen (NO_3_^−^-N) and ammonium nitrogen (NH_4_^+^-N) were determined by a spectrophotometry method [23]. Soil microbial biomass carbon (MBC), nitrogen (MBN), and phosphorus (MBP) were determined using the chloroform fumigation extraction method [24].

### 2.4. DNA Extraction and PCR Amplification

Soil microbial DNA was extracted three times from each soil sample (0.5 g) using the Fast soil DNA SPIN Kit (MP Biomedicals, Santa Ana, CA, USA). The extracted soil DNA was diluted in 50 μL of sterilized water. Subsequently, the concentration of the extracted DNA was determined using a NanoDrop ND-1000 spectrophotometer (Nanodrop Technologies, Wilmington, DE, USA). The DNA samples were stored at −80 °C for further analysis.

The hypervariable region V3–V4 of the bacterial 16S rRNA gene and the internal transcribed spacer (ITS) regions of fungal gene were amplified using the primers 338F/806R [25] and ITS1F/ITS2R [26], respectively. The bacterial amplification solution consisted of 4 μL FastPfu Buffer (5×), 2 μL dNTPs (2.5 mM), 0.8 μL each of Forward Primer (5 μM) and Reverse Primer (5 μM), 0.4 μL FastPfu Polymerse (Beijing TransGen Biotech Co., Ltd., Beijing, China), 0.2 μL BSA, 10 ng Template DNA, and ddH_2_O to a final volume of 20 μL. The amplification protocol involved denaturation at 95 °C for 3 min, followed by 27 cycles of denaturation at 95 °C for 30 s, annealing at 55 °C for 30 s, and extension at 72 °C for 45 s, with a final extension at 72 °C for 10 min. For the ITS rRNA qPCR reaction, a 20 μL mixture was prepared containing 2 μL of Buffer (10×), 2 μL dNTPs (2.5 mM), 0.8 μL each Forward and Reverse Primers (5 μM), 0.2 μL rTaq Polymerse (Shanghai Fusheng Industrial Co., Ltd., Shanghai, China), 0.2 μL BSA, 10 ng Template DNA, and ddH_2_O. The amplification protocol was similar to the bacterial, but with 35 cycles and a final holding at 10 °C until halted.

Each sample was run in triplicate on the ABI GeneAmp 9700 system (Applied Biosystems, Waltham, MA, USA). The relative amplicons were then pooled to create the final PCR product. Each mixed gene sample (i.e., 16S rRNA gene and ITS rRNA gene) underwent electrophoresis on 2% agarose gel. DNA fragments of the expected size (301–400 bp for 16S rRNA gene and 201–300 bp for ITS rRNA gene) were recovered using an AxyPrep DNA Gel Recovery Kit (Axygen Biosciences (Hangzhou) Co., Ltd., Hangzhou, China). Finally, the amplicon libraries were sequenced on the Illumina’ HiSeq 2000 platform (Illumina, San Diego, CA, USA) at Majorbio Bio-Pharm Technology Co., Ltd. (Shanghai, China).

### 2.5. Bioinformatics Processing

The raw gene sequencing reads were demultiplexed, quality-filtered, and then merged using QIIME [27] based on the three criteria [28]. The operational taxonomic units (OTUs) were clustered at a similarity level of 97% using Uparse v7.0.1 [29], after chimeric sequences were removed. Taxonomy identification of the OTUs was performed using a naïve Bayesian classifier algorithm implemented in mother [30,31], against the Silva v138/16s_bacteria database for 16S rRNA and the unite 7.2/its_fungi database for an ITS rRNA, at a 0.7 confidence threshold. The complete datasets were deposited in the Sequence Read Archive (SRA) database of the National Center for Biotechnology Information (NCBI) with the accession numbers of PRJNA 898882 for bacteria and PRJNA 899297 for fungi.

### 2.6. Statistical Analysis

To measure the α diversity of the soil microbial community, we utilized the Shannon–Wiener index [32] and observed richness, defined as the total number of OTUs normalized to a specific number of reads per sample. To assess the β diversity among the soil microbial communities across different forest succession stages, non-metric multidimensional scaling (NMDS) was conducted based on the Bray–Curtis distance. The quality of NMDS fit was assessed using stress values, where stress > 0.2 indicates poor goodness of fit, 0.1–0.2 indicates fair goodness of fit, 0.05–0.1 suggests good fitness, and stress < 0.05 denotes excellent fitness [33]. Additionally, the Wilcoxon rank sum test was used to analyze differences in soil microbial diversity indices [34]. To identify the dominant factors influencing bacterial and fungal community composition, redundancy analysis (RDA) based on Bray–Curtis distance was conducted [35]. Furthermore, variation partition analysis was performed to determine the contribution of soil, vegetation, and microbial biomass to soil microbial community variance [36]. All these analyses were conducted using R 3.3.4 [37]. To construct the co-occurrence network of the bacterial and fungal community, we considered the taxa with >1% relative abundance at the genus level in the three forests and calculated Pearson correlations [38,39]. Only edges with Pearson correlations exceeding 0.7 and adjusted *p*-values below 0.05 were retained for the network visualization, which was created using Gephi 0.9.2.

## 3. Results

### 3.1. Plants and Soil Properties During Karst Forest Succession

The average DBH, richness, and Shannon–Wiener index of woody plants increase with forest succession, from shrubland to secondary forest, and finally to primary forest. Notably, all three properties of woody plant in the primary forest are significantly higher compared to those in the shrubland (*p* < 0.05) (Table 1). The soils in both the shrubland and primary forest exhibit a high organic carbon content, exceeding 30 g·kg^−1^ (Table 1). However, soil properties vary significantly along karst forest succession, with the primary forest generally exhibiting the highest nutrient levels (Table 1).

### 3.2. Dynamics in Bacterial and Fungal Community Composition and Diversity During Forest Succession

*Proteobacteria*, *Actinobacteria*, *Acidobacteria*, and *Chloroflexi* are the dominant bacterial phyla in the karst forest soils, collectively accounting for 79–81% of the total bacterial abundance (Figure 1a). Meanwhile, *Ascomycota* and *Basidiomycota* are the primary fungal phyla in these forests, with a relative abundance ranging from 76% to 81% (Figure 1b). Regarding fungal α-diversity, based on richness and Shannon–Wiener index, there is an initial decrease followed by an increase during karst forest succession. However, there are no significant differences among the three forests (Figure A2b,d). In contrast, the soil bacterial richness at the phylum level is significantly higher in the secondary forest compared to both the shrubland and primary forest (Figure A2a). Additionally, the Shannon–Wiener index for soil bacteria at the phylum level is significantly lower in the primary forest than in both the shrubland and secondary forest (Figure A2c). The stress values obtained from NMDS analysis (0.0801 for bacteria and 0.153 for fungi) indicate the good fitness of the models. The NMDS results reveal that the soil bacterial community in the primary forest is closely related to that in the shrubland but distinct from that in the secondary forest (Figure 2a, Adonis, *R*^2^ = 0.407, *p* = 0.001). Conversely, the fungal communities among the three forests can be discriminated (Figure 2b, Adonis, *R*^2^ = 0.194, *p* = 0.001).

### 3.3. Co-Occurrence Networks for Soil Bacteria and Fungi in the Karst Forest Succession

Co-occurrence networks for soil bacteria and fungi in the three karst forests (Figure 3) are constructed based on abundant taxa with a relative abundance >1% at the genus level. The bacterial network comprises 124 edges and 37 nods, while the fungal network has 24 edges and 17 nodes (Table A1). In the bacterial co-occurrence network across the three karst forests, JG30-KF-CM66, Subgroup_17, Subgroup_9, *Anaerolineae*, *Dehalococcoidia*, Gitt-GS-136, *Gemmatimonadetes*, and *Chloroflexia* are identified as the most important nodes (Figure 3a). Conversely, for fungi, the key nodes in the forests include *Sordariomycetes*, *Mortierllomycetes*, *Eurotiomycetes*, *Leotiomycetes*, GS-14, and *Kickxellomycetes* (Figure 3b). Notably, the bacterial networks in the karst forest exhibit a higher number of nodes, edges, and average path compared to the fungal networks (Figure 3 and Table A1).

### 3.4. Drivers Regulating Soil Bacterial and Fungal Profiles Among Karst Forests

The db-RDA results show that soil pH, SOC, Total N, MBP, moisture, MBN, AK, and the Shannon index of woody plants significantly affect the soil bacterial community in the karst forests (Figure 4a, Table A2). In contrast, only soil pH significantly affects the soil fungal community in the karst forests (Figure 4b, Table A2). Further variation partition analysis reveals that soil microbial biomass accounts for 7.6% of the variation in the soil bacterial community. Soil properties and plant factors explains 4.1% and 3.4% of the variation in the soil bacterial community, respectively (Figure 5a). For the soil fungal community, soil properties account for 9.3% of the variation (Figure 5b). Woody plants have minimal impact on the soil fungal community (Figure 5b). Notably, a large proportion of the variation remains unexplained for both the soil bacterial and fungal communities, accounting for 82.0% and 74.8%, respectively (Figure 5).

## 4. Discussion

### 4.1. Dynamics in Soil Microbes and Environmental Variables Along Karst Forest Succession

The implementation of ecological engineering, including natural restoration, has been widespread adopted worldwide [40]. For instance, in a karst region of southwest China [14,16], natural restoration has undoubtedly enhanced carbon sequestration through accumulation in biomass and soil organic carbon [41]. Additionally, without human or animal disturbance, plant natural succession alters vegetation growth, community composition, and productivity [21]. These changes, in turn, impact the belowground status and functions [42,43]. In our study, *Proteobacteria*, *Actinobacteria*, and *Acidobactria* dominate the bacterial community at the phylum level (Figure 1a), while *Ascomycota* and *Basidiomycota* dominate the fungal communities (Figure 1b), regardless of the forest succession stage in the karst region. This finding aligns with previous studies conducted in karst regions [19,44], indicating that forest succession had no significant effects on the dominant soil bacterial and fungal community structure. It is worth noting that the *Proteobacteria* communities are higher in the secondary forest compared to the shrubland (Figure 1a). This suggests that soil conditions improved during early plant succession by favoring *Proteobacteria* in a copiotrophic environment with available labile substrates [45]. Similar observations have been made in other studies. For example, *Protebacteria* communities increased with secondary succession after abandonment in the Loess Plateau in China [8] and with vegetation succession along the Franz Josef chronosequence in New Zealand [46].

We find that the bacterial α-diversity is lowest in the primary forest (Figure A2a,c), despite the high diversity of woody plants listed in Table 1. This finding contrasts with the commonly held belief that plant diversity is positively related to soil bacterial diversity [47,48]. This contradiction might be explained by the differences in the substrates availability between our study’s forest ecosystems and the grassland ecosystems previous examined [47,48]. Furthermore, fungal α-diversity followed an initial downward and then upward trend during karst forest succession (Figure A2b,d). These findings suggest that the fungal and bacterial communities respond differently to forest succession [19]. The distinct responses may be attributed to how these communities respond differently to changing soil properties during forest succession [49].

### 4.2. Divergent Patterns of Bacteria and Fungi Along Karst Forest Succession

The NMDS results show that the compositions of soil bacterial and fungal communities in primary forest exhibit greater similarity to those in shrubland, in contrast to secondary forest (Figure 2). The phenomenon may be caused by two factors. Firstly, the soil physical and chemical properties under shrubland and primary forest are comparable (Table 1), fostering a similar environment for microbial survival and proliferation. Secondly, evergreen tree species prevalent in primary forests possess higher C:N ratios than deciduous tree species, stemming from their unique woody plant composition [20,50,51]. Although microorganisms may prefer the litter of evergreen trees, their litter production is relatively scant compared to deciduous tree species, resulting in a microbial community composition that mirrors that of shrublands with a higher proportion of deciduous trees [50]. However, in subtropical non-karst regions, soil bacterial communities vary with forest succession, potentially due to the increased production and accumulation of bacterial residues as succession progressed [52]. The divergence suggests that karst forests may undergo a more intricate succession process compared to non-karst forest. Generally, soil fungal communities undergo changes during forest succession (Figure 2b), indicating that specific fungal species or taxa fluctuate with successional stage. For instances, arbuscular mycorrhizal (AM) fungi tend to be earlier colonizers of successional habitats [53]. The results from the microbial co-occurrence network (Figure 3 and Table A1) indicate that, despite their lower abundance, the correlations among soil fungi in forests are notably stronger than those observed among bacteria. This may be due to the more pronounced and enduring interactions between various fungal species [19], such as the symbiotic relationship between diazotrophs and arbuscular mycorrhizal fungi [54]. Our findings further imply that the diversity of bacteria and fungi across forest succession stages in karst regions may surpass that observed in non-karst regions, such as the Loess Plateau [49].

Researchers have emphasized that the prevalence of highly interconnected taxa, such as kinless hubs, within soil microbial networks correlate with elevated functional potential in terrestrial ecosystems [55]. Our study’s co-occurrence network reveals that JG30-KF-CM66, Subgroup_17, Subgroup_9, *Anaerolineae*, *Dehalococcoidia*, Gitt-GS-136, *Gemmatimonadetes*, and *Chloroflexia* are the most prominent bacterial nodes. Meanwhile, *Kickxellomycetes* stands out as the most significant fungal node (Figure 3b). These nodes exhibit robust connection with other taxa within the network, indicating a strong link to functional potential. Notably, JG30-KF-CM66 has been documented to play a role in global cobalamin production via the cobinamide to cobalamin salvage pathway [56]. Additionally, *Sordariomycetes*, composed of typical *saprotrophic* fungi, excel at decomposing labile C [57], whereas *Mortierllomycetes* show a robust response to readily degradable, N-rich substrates [58]. Therefore, the observed co-occurrence patterns indicate that species interactions play a more pivotal role in soil nutrient processes or functions compared to microbial diversity [19].

### 4.3. Drivers of Soil Microbial Dynamics Along Karst Forest Succession

Bacterial community structure along vegetation succession is predominately influenced by variations in soil nutrients, plant diversity, and composition [8,21]. Our analysis, utilizing db-RDA and variance partition analysis, has revealed that soil properties such as pH, SOC, Total N, moisture, and AK, along with soil microbial biomass P and N, and woody plant diversity are the key factors driving soil bacterial community dynamics during karst forest succession (Figure 4a and Figure 5a, Table A2). Our findings further indicate that the diversity and biomass of woody plants, which are highly positively related with DBH [59], can negatively impact soil bacterial diversity during forest succession (Table A2). This may explain why secondary forests, despite having lower SOC and soil N content compared to other forests (Table 1), harbor the most abundant bacteria (Figure A2), and why primary forests, with the highest available K (Table 1), exhibit lower bacteria richness (Figure A2). Our observation contrasts with studies conducted in undisturbed grasslands [47] and during plant secondary succession following abandoned farmland [8,48], where plant diversity is positively associated with soil bacterial diversity. This finding indicates that the relationship between plant diversity and bacterial diversity evolves across different succession stages. One plausible explanation for this shift is that in undisturbed grasslands or during the early succession stages, plant diversity provides a diverse range of niches for bacteria colonization and growth. However, in the late succession stage, such as forest, woody plant biomass, indicated by DBH [59], exhibits a significant negative correlation with soil bacterial diversity (Table A3). This implies that higher woody plant diversity and biomass may promote plant–soil feedback that enhances the stability of soil bacteria rather than the diversity [42]. Such feedback may include nutrient cycling and modulation of the soil microenvironment, which can indirectly influence bacterial community structure and function [42].

As forest succession advances, plant regeneration occurs and exerts a notable influence on soil properties, including pH, organic inputs, and available nutrients (Table 1). Notably, there are significant correlations between SOC, total N, AN, NO_3_^−^-N, and MBN with soil bacterial alpha diversity (Table A3). This underscores the pivotal role of soil carbon and nitrogen in shaping bacterial diversity [60]. Intriguingly, while NH_4_^+^-N does not exhibit a significant correlation with the relative abundances of dominant bacterial phyla, NO_3_^−^-N does (Figure 4a, Table A3). This finding aligns with previous research suggesting that variations in bacterial communities during forest succession can be attributed to different N fractions [61]. The distinct relationship between NH_4_^+^-N and NO_3_^−^-N with bacterial abundances highlights the intricate interplay between nitrogen availability and bacterial dynamics during forest succession.

It has been established that plant diversity exerts a global influence on soil fungal communities [61], a finding that is corroborated by research examining plant secondary succession on the Loess Plateau in China [21]. This influence primarily arises from the diverse range of food resources that plants provide to fungi, encompassing root exudates and litter [62,63]. In particular, fungal groups such as *Ascomycota* and *Basidiomycota* play crucial roles in the decomposition and rhizodeposition of organic substrates [62,64]. In our study, we observed that soil pH is the primary factor explaining the response of the soil fungal community to forest succession (Figure 4b and Figure 5b, Table A2). This finding contrasts with the results showing that shrubland and primary forest have similar pH values (Table 1), yet there is considerable variation in fungal richness (Figure A2). This discrepancy may be attributed to the higher tree species richness in primary forest (Table 1), as many ectomycorrhizal fungi belong to the phyla of *Ascomycota* and *Basidiomycota* identified in our study. Furthermore, *Ascomycota* and *Basidiomycota* show significant correlations with most soil properties, but not with woody plant diversity (Table A3). This indicates that fungal community compositions, particularly the dominant phyla *Ascomycota* and *Basidiomycota*, respond significantly to forest succession, which is dependent on soil pH, C, and N dynamics. This phenomenon can be attributed to the dynamics of soil properties during forest succession in the subtropical climate and unique karst habitat. The karst terrain, characterized by its distinct geology and hydrology [16], combined with the subtropical climate, creates a unique environment that shapes soil properties and nutrient cycling. These conditions, in turn, directly affect fungal communities, which heavily rely on soil resources and conditions for their growth and reproduction. The significant role of soil pH, C, and N in determining fungal community structure and diversity underscores the importance of considering soil properties when studying fungal ecology in forest ecosystems.

## 5. Conclusions

Forest succession exerts diverse influence on soil bacterial and fungal diversity, community composition, and co-occurrence patterns. Specifically, bacterial diversity in secondary forest significantly differs from that in shrubland and primary forest, whereas fungal diversity is distinctly differentiated across the three stages of karst forest succession. The co-occurrence patterns suggest that soil fungi exhibit fewer but more intense relationships compared to bacteria in these karst forests, indicating that bacteria and fungi adopt distinct strategies during forest succession. Furthermore, soil properties, including pH, SOC, Total N, AK, MBP, and MBN, along with woody plant diversity, collectively influence the bacterial community. Among these, soil properties, particularly pH, are the most dominant factor controlling the fungal community. Variations in soil nutrient status can effectively predict the changes in soil bacterial and fungal community composition and diversity throughout forest succession. In summary, intermediate pH levels and nutrient-rich soil conditions favor both bacterial and fungal communities and their interactions in karst forests. Therefore, understanding the intricate interactions between soil properties, plant diversity, and microbial communities is crucial for comprehending the ecological processes that shape forest ecosystems.

## Figures and Tables

**Figure 1 microorganisms-12-02136-f001:**
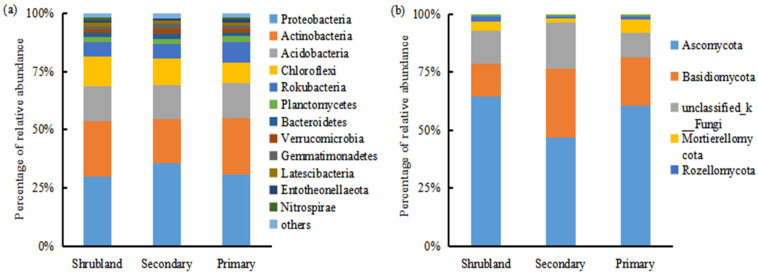
Soil bacterial (**a**) and fungal (**b**) community composition at the phylum level along forest succession from shrubland and secondary forest to primary forest.

**Figure 2 microorganisms-12-02136-f002:**
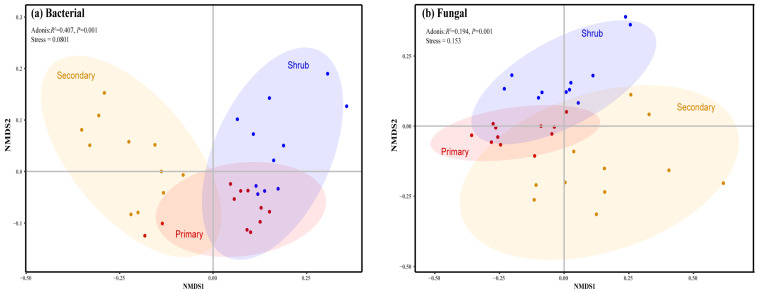
NMDS of soil bacterial (**a**) and fungal (**b**) communities along forest succession from shrubland and secondary forest to primary forest. Green circles indicate soil samples in the shrubland; orange circles indicate soil samples in the secondary forest; and red circles indicate soil samples in the primary forest.

**Figure 3 microorganisms-12-02136-f003:**
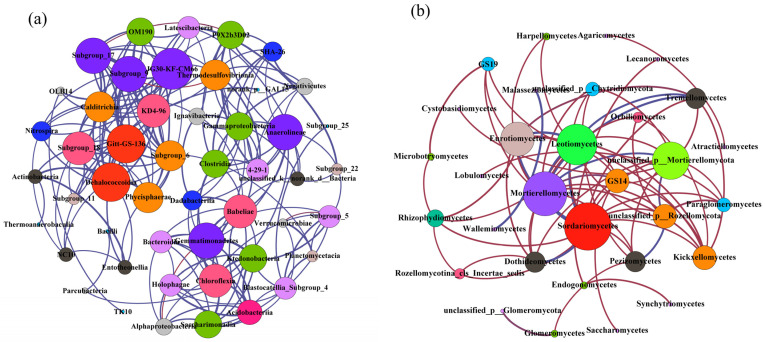
The co-occurrence network of soil microbial organisms in the three forests. (**a**) Bacterial co-occurrence network and (**b**) fungal co-occurrence network.

**Figure 4 microorganisms-12-02136-f004:**
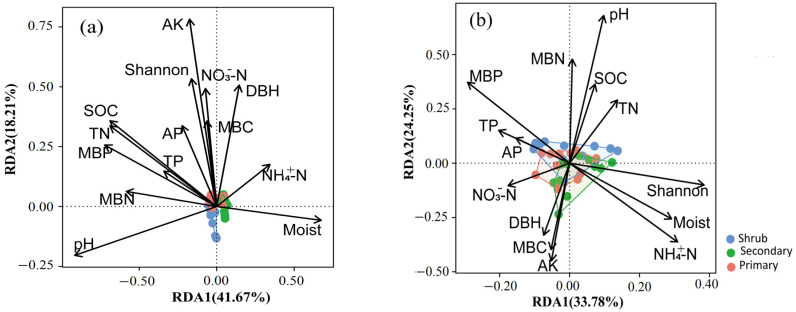
Ordination diagram of db-RDA with soil microbial community composition and environmental factors. (**a**) Bacterial community composition; (**b**) fungal community composition.

**Figure 5 microorganisms-12-02136-f005:**
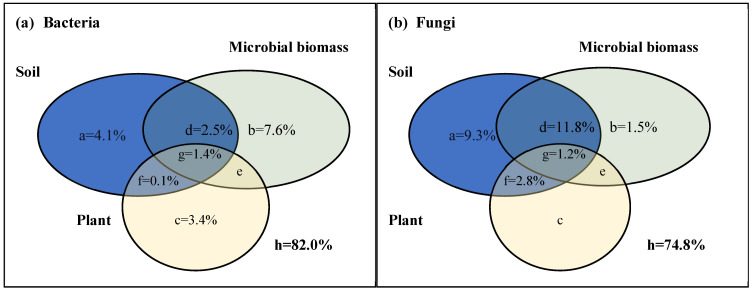
The variation partition of environmental factors for soil microbial community in the karst forests. (**a**) Bacteria, bacterial community; (**b**) fungi, fungal community; a, b, and c, represented pure interpretation from soil, microbial biomass, and plant, respectively; d, soil and microbial biomass joint interpretation; e, microbial biomass and plant joint interpretation; f, soil and plant joint interpretation; g, soil, microbial biomass, and plant joint interpretation; h, residuals, the unexplained part. The letters without values indicated the values were less than 0.

**Table 1 microorganisms-12-02136-t001:** Woody plant and soil properties during karst forest succession (*n* = 11, means ± SE).

Properties		Shrubland	Secondary Forest	Primary Forest
Woody plants	Average DBH (cm)	3.6 ± 0.4 b	4.9 ± 0.6 ab	5.4 ± 0.1 a
	Richness	8.7 ± 1.2 b	16.1 ± 2.8 b	29.4 ± 4.2 a
	Shannon-Wiener index	1.2 ± 0.2 b	1.9 ± 0.3 ab	2.2 ± 0.3 a
Soil properties	SOC(g·kg^−1^)	30.3 ± 1.6 a	19.8 ± 2.5 b	35.6 ± 2.8 a
	Total N (g·kg^−1^)	5.6 ± 0.4 a	4.0 ± 0.4 b	6.2 ± 0.5 a
	Total P (g·kg^−1^)	1.14 ± 0.1 a	0.97 ± 0.1 b	1.2 ± 0.0 a
	Total K (g·kg^−1^)	5.8 ± 0.7 b	14.9 ± 0.7 a	5.4 ± 0.7 b
	AN (mg·kg^−1^)	250.9 ± 26.1 a	215.0 ± 32.2 b	307.3 ± 22.6 a
	AP (mg·kg^−1^)	3.0 ± 0.3 b	3.3 ± 0.6 ab	4.6 ± 0.4 a
	AK(mg·kg^−1^)	58.4 ± 5.8 b	74.3 ± 3.0 ab	76.8 ± 5.7 a

Different letters in the same row indicate significant differences between the karst forests (*p* < 0.05).

## Data Availability

Data is available if requested.

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
