# Peer review of "Soil Properties Regulate Soil Microbial Communities During Forest Succession in a Karst Region of Southwest China"

_microorganisms, 2024, doi:10.3390/microorganisms12112136_

Round 1

Reviewer 1 Report

Comments and Suggestions for Authors

1. The materials and methods present results from the literature. This should be moved to the results and discussion section. e.g The 91 dominant woody vegetation in the shrubland plot in Mulian are Vitex negundo, Alangium 92 chinense, and Ligustrum japonicum [22] (lines 91-92).

2. A major disadvantage of the work is the lack of comparison of the results to the current state of knowledge.

3. A major disadvantage of the work is also the lack of demonstration of the novelty aspect. An environmental analysis has been carried out and what further benefits does it bring?

Comments on the Quality of English Language

Repeating the same phrases - requires improvement:

Belowground biodiversity, especially microbial diversity, plays a crucial role in reg-35 ulating aboveground biodiversity and ecosystem functioning [1], especially microbial di-36 versity could drive multifunctionality in terrestrial ecosystems, including climate buffer, 37 soil fertility, and material production and provision [2-3],

Author Response

Comment 1The materials and methods present results from the literature. This should be moved to the results and discussion section. e.g The dominant woody vegetation in the shrubland plot in Mulian are Vitex negundo, Alangium chinense, and Ligustrum japonicum [22] (lines 91-92).

Response: Thank you for reviewing my paper and providing valuable feedback. Regarding the suggestion to move the information 'The dominant woody vegetation in the shrubland plot in Mulian are Vitex negundo, Alangium chinense, and Ligustrum japonicum [22]' to the 'Results and Discussion' section, after careful consideration, we believe this information is more appropriately placed in the 'Materials and Methods' section. This information describes the basic vegetation of the study site (the shrubland plot in Mulian), which is fundamental to the research design and sample selection. Understanding the dominant woody vegetation of the site is crucial for readers to comprehend the experimental background, ecological characteristics of the study location, and potential factors that may influence the experimental results. Therefore, we maintain that retaining this information in the 'Materials and Methods' section will more effectively provide readers with relevant information about the research background and experimental design. Additionally, I recognize the importance of ensuring the accuracy and logical flow of information when writing a paper. Thus, we will more clearly indicate the role and significance of this information in the 'Materials and Methods' section to enhance the structure and readability of the paper. Thank you again for your attention and constructive comments.

Comment 2A major disadvantage of the work is the lack of comparison of the results to the current state of knowledge.

Response: Thanks for your constructive comments, and we have thoroughly checked the manuscript. Based on our results, the comparisons within the same region and with other ecosystems were conducted in the discussion. For instance, the comparison mentioned in Lines 280-292, Lines 331-333 specifically address this point.

Lines 280-292: In our study, Proteobacteria, Actinobacteria, and Acidobactria dominate the bacterial community at the phylum level (Figure 1a), while Ascomycota and Basidiomycota dominate the fungal communities (Figure 1b), regardless of the forest succession stage in the karst region. This finding aligns with previous studies conducted in karst regions [19, 44], indicating that forest succession had no significant effects on the dominant soil bacterial and fungal community structure. It is worth noting that the Proteobacteria communities are higher in the secondary forest compared to the shrubland, (Figure 1a). This suggests that soil conditions improved during early plant succession by favoring Proteobacteria in a copiotrophic environment with available labile substrates [45]. Similar observations have been made in other studies. For example, Protebacteria communities increased with secondary succession after abandonment in the Loess Plateau in China [8] and with vegetation succession along the Franz Josef chronosequence in New Zealand [46]. 

Lines 331-333: Our findings further imply that the diversity of bacteria and fungi across forest succession stages in karst regions may surpass that observed in non-karst regions, such as the Loess Plateau [49].

Comment 3A major disadvantage of the work is also the lack of demonstration of the novelty aspect. An environmental analysis has been carried out and what further benefits does it bring?

Response: Thanks for your constructive comments. Indeed, a significant drawback of our work lies in the insufficient demonstration of its novelty. Our innovation specifically focuses on the dynamic changes of soil bacterial and fungal communities and their primary regulatory factors across different stages of forest succession in karst areas. We discovered that in karst forest ecosystems, soil physicochemical properties play a crucial role in regulating these microbial communities. Specifically, as concluded in Lines 423-425: “In summary, intermediate pH levels and nutrient-rich soil conditions favor both bacterial and fungal communities and their interactions in karst forests.” Additionally, in the discussion section, we have emphasized the analysis of similarities and differences between non-karst areas or other terrestrial ecosystems in the discussion section. To highlight these relevant changes, we have marked them in red font.

Comment 4: Comments on the quality of English language: Repeating the same phrases-requires improvement: Belowground biodiversity, especially microbial diversity, plays a crucial role in regulating aboveground biodiversity and ecosystem functioning [1], especially microbial diversity could drive multifunctionality in terrestrial ecosystems, including climate buffer, soil fertility, and material production and provision [2-3],

Response: Thanks for you constructive comments. We rewrote this sentence in Line 35-40:

Belowground biodiversity, notably microbial diversity, plays a crucial role in regulating aboveground biodiversity and ecosystem functioning [1]. Furthermore, microbial diversity is a key driver of multifunctionality in terrestrial ecosystems, encompassing critical aspects such as climate regulation, soil fertility, material production, and provision [2-3]. These contributions ultimately enhance human well-being [4].

Reviewer 2 Report

Comments and Suggestions for Authors

Dear Editor

Many thanks for considering me as a potential reviewer for the article " Soil Properties Regulate Soil Microbial Communities during Forest Succession in a Karst Region of Southwest China" The article is undoubtedly well-structured, well-presented and well-written. However, I have several observations that should be considered before proceeding further.

My observations are as follows.

Major considerations

·       Please authorize all plant names i.e. Vitex negundo, Alangium chinense, and Ligustrum japonicum and soo on…. You can use WFO plant list: https://wfoplantlist.org/taxon/wfo-0000333303-2024-06?page=1... Do this throughout the manuscript.

·       Your introduction should be improved thereby adding information regarding various physio-chemical parameters you took under consideration in the study, please give some examples and case studies to strengthen and justify your findings.

·       Please refine your aim/objectives.

·       Please split this longer sentence ‘Belowground biodiversity, especially microbial diversity, plays a crucial role in regulating aboveground biodiversity and ecosystem functioning [1], especially microbial diversity could drive multifunctionality in terrestrial ecosystems, including climate buffer, soil fertility, and material production and provision [2-3], and ultimately improve human 38 well-being [4].’

·       Lines 73-83; I don’t know if this paragraph should be in the introduction. I would like it to be in the material and method section….please have a look,

·       Lines 56-57 (Recent researches have demonstrated) and 62-65 (Some other studies); authors claimed about studies but I did not see their citations, please do consider this issue.

·       Sub-section 2.3. Soil Sample Collection and Determination; Determination what?

·       Line 133, where is ‘NO3 -N and NH4’ full form?,

·       P<0.05…p should be in the smaller letter, please do consider throughout the manuscript,

·       Tables and figures are bold in the descriptions, while in the texts its not, please create unity,

·        Line-301 for please do format to Time new Roman

Author Response

Comment 1: Please authorize all plant names i.e. Vitex negundo, Alangium chinense, and Ligustrum japonicum and so on…. You can use WFO plant list: https://wfoplantlist.org/taxon/wfo-0000333303-2024-06?page=1... Do this throughout the manuscript.

Response: Thanks for your valuable comments. We have checked all plant names throughout the manuscript using WFO plant list (https://wfoplantlist.org/) and highlighted the revised names in red font in Line 95-102:

The dominant woody vegetation in the shrubland plot in Mulian are Vitex negundo L., Alangium chinense (Lour.) Harms, and Ligustrum quihoui Carrière [22], which naturally recovered from abandonment in 1985 to present. Bauhinia brachycarpa Thunb., Cipadessa baccifera Miq., Radermachera sinica (Hance) Hemsl., and Toona sinensis (A.Juss.) M.Roem. are the dominant species in the secondary forest plot in Guzhou with less disturbance due to the application of projects of comprehensive control for rocky desertification [22]. The dominant species in the primary forest in Mulun are Cryptocarya concinna Hance, Itoa orientalis Hemsl., and Brassaiopsis glomerata (Blume) Regel [22]. 

Comment 2: Your introduction should be improved thereby adding information regarding various physio-chemical parameters you took under consideration in the study, please give some examples and case studies to strengthen and justify your findings.

Response: Thank you for your valuable feedback. We appreciate your suggestion to enrich my introduction by incorporating more details about the various physicochemical parameters considered in the study, specifically in Lines 43-50. Here’s the revised version:

Various environmental factors play regulatory roles in shaping soil microbial diversity. For instance, temperature and soil carbon content influence soil archaea, while aridity, vegetation characteristics, and soil pH regulate bacterial communities, as reported in [8]. During secondary succession, the soil bacterial community is affected by multiple factors, including plant diversity and composition, as well as soil nutrients such as total organic carbon and total nitrogen, as detailed in [9]. Additionally, soil fungal diversity and functionality are influenced by plant species during afforestation processes [10]. 

Comment 3:  Please refine your aim/objectives.

Response: Thank you for your feedback. I understand the importance of refining my aim/objectives to ensure they are clear, specific, and achievable. Upon reflection, I have revised my aims/objectives in Lines 75-80:

To bridge this knowledge gap, we conducted a study in a karst region of Southwest China. In this study, we employed sequencing of 16S rRNA gene and ITS gene amplicons to gain insights into the community composition and diversity of soil bacteria and fungi across various stages of restoration, including shrubland, secondary forest, and primary forest. Subsequently, we quantified the soil microbial community, its diversity, and the co-occurrence network in relation to karst forest succession. Our ultimate goal was to identify the key factors influencing soil microbial dynamics during the process. This research contributes to a deeper understanding of the critical role that soil microorganisms play in maintaining ecosystem functions and services in typical karst areas. 

Comment 4: Please split this longer sentence ‘Belowground biodiversity, especially microbial diversity, plays a crucial role in regulating aboveground biodiversity and ecosystem functioning [1], especially microbial diversity could drive multifunctionality in terrestrial ecosystems, including climate buffer, soil fertility, and material production and provision [2-3], and ultimately improve human well-being [4].’

Response: We accepted your constructive suggestion and we rewrote this sentence in Line 35-40: Belowground biodiversity, notably microbial diversity, plays a crucial role in regulating aboveground biodiversity and ecosystem functioning [1]. Furthermore, microbial diversity is a key driver of multifunctionality in terrestrial ecosystems, encompassing critical aspects such as climate regulation, soil fertility, material production, and provision [2-3]. These contributions ultimately enhance human well-being [4].

Comment 5: Lines 73-83; I don’t know if this paragraph should be in the introduction. I would like it to be in the material and method section….please have a look,

Response: Thank you for your feedback. Regarding the paragraph on lines 73-83, I understand your concern about its placement. While the content of this paragraph does offer a straightforward description of a specific step or process within the experiment or study, it was originally intended to provide context for our research based on the background and aims outlined in the first and second paragraphs of the Introduction. Upon further reflection, we acknowledge that placing this paragraph directly in the 'Materials and Methods' section might not be entirely appropriate. However, we have taken your comments into consideration and re-examined the content. To maintain coherence, we have decided to remove the part that describes materials and methods specifically, ensuring that the remaining content aligns better with the Introduction section while avoiding redundancy.

Comment 6:   Lines 56-57 (Recent researches have demonstrated) and 62-65 (Some other studies); authors claimed about studies but I did not see their citations, please do consider this issue.

Response: Thank you for your thorough reminder. We have incorporated the citations at the end of the relative sentences. following your suggestion, we have appended the respective citations after the ‘Recent researches’ and ‘Some other studies’, respectively.

Comment 7: Sub-section 2.3. Soil Sample Collection and Determination; Determination what?

Response: In the sub-section 2.3, “Soil Sample Collection and Determination”, the focus of the determination primarily encompass various aspects, including the physicochemical properties of the soil, nutrient contents, and soil microbial biomass. Furthermore, a detailed description of determination methods is provided in Lines 129-139.

Soil pH was determined by a pH meter using 1:5 soil/water suspension. The determination of soil organic carbon (SOC), total nitrogen (Total N), and total phosphorus (Total P) adopted the methods of Walkley-Black wet oxidation, semi-micro Kjeldahl, alkali digestion-molybdenum antimony colorimetry, and flame photometry, respectively [23]. The determination of available nitrogen (AN), available phosphorus (AP), and available potassium (AK) adopted the alkaline hydrolysis diffusion method, Olsen method, and OAc method, respectively [23]. The exchangeable Ca2+ and Mg2+ were determined by EDTA titration method, and nitrate nitrogen (NO3--N), and ammonium nitrogen (NH4+-N) were determined by spectrophotometry method [23]. Soil microbial biomass carbon (MBC), nitrogen (MBN), and phosphorus (MBP) were determined using the chloroform fumigation-extraction method [24].

Comment 8Line 133, where is ‘NO3 -N and NH4’ full form?,

Response: We apologize for the unclear description earlier. In Line 133, 'NO3--N and NH4+-N' refer to 'Nitrate Nitrogen' and 'Ammonium nitrogen', respectively. Specifically, NO3--N is used to describe the nitrogen content in water bodies or soils in the form of nitrate, while NH4+-N typically represents ammonium nitrogen, which play a significant role in the nitrogen cycle. To address the clarity issue, we revised ‘NO3 -N and NH4’ to “nitrate nitrogen (NO3--N), and ammonium nitrogen (NH4+-N)” in Line 139-140.

Comment 9: P<0.05…p should be in the smaller letter, please do consider throughout the manuscript,

Response: We accepted your suggestion and revised ‘P<0.05’ to ‘p<0.05’ and ‘P=0.001’ to ‘p=0.001’throughout the entire manuscript for consistency and to adhere to standard statistical notation, respectively.

Comment 10: Tables and figures are bold in the descriptions, while in the texts its not, please create unity,

Response: Thank you for your feedback regarding the inconsistency where tables and figures have bold descriptions but the corresponding text does not. We have taken note of this issue and have made adjustments by using bold font in the text to ensure unity and professionalism throughout the document.

Comment 11: Line-301 for please do format to Time new Roman

Response: Thank you very much for your careful reminder. We changed the ‘for’ to Time New Roman in Line 301 (now Line 313 in the revised manuscript).

Round 2

Reviewer 2 Report

Comments and Suggestions for Authors

Dear Editor/Authors,

Many thanks for considering the updated and revised version of the article.

The article is much more refined, however, there are still some improvements, that should be taken into consideration. My queries are,

·       Line-28, whereas soil properties (e.g., pH), why properties and only one example? Why not directly ‘whereas soil pH….

·       Line-75, please cite your studies, To bridge this knowledge gap, we conducted a study in a karst region of Southwest China.

·       Line 918, please pay attention P values below 0,05, p should be smaller and 0,05 should be 0.05,

·       Figure 1, the legend (a) and other indications (writing), seem blurred, compared to Figure b,

·       In general, the sentences (throughout the manuscript) seem longer, please pay attention to it (my personal suggestion).

Comments on the Quality of English Language

Dear Editor/Authors,

A few sentences are still confusing, I think, minor English editing is recommended.

Thanks

Author Response

Comment 1:  Line-28, whereas soil properties (e.g., pH), why properties and only one example? Why not directly ‘whereas soil pH….

Response: We acknowledge the suggestion and have revised the sentence to directly mention soil pH, as highlighted in red font in our revision.

Comment 2: Line-75, please cite your studies, To bridge this knowledge gap, we conducted a study in a karst region of Southwest China.

Response: Thank you for your reminder regarding Line 75. To address the knowledge gap discussed in the sentence, "To bridge this knowledge gap, we conducted a study in a karst region of Southwest China," we have now included the relevant references [7, 18, 19, 21] in the manuscript. These references provide background information and support our claim about the existing knowledge gap, as well as the necessity and relevance of our study conducted in the karst region of Southwest China. We apologize for any confusion and appreciate your constructive feedback.

Comment 3: Line 918, please pay attention P values below 0,05, p should be smaller and 0,05 should be 0.05

Response: Sorry for the mistake in the manuscript and we revised it in red fond.

Comment 4:  Figure 1, the legend (a) and other indications (writing), seem blurred, compared to Figure b.

Response:  Thank you for bringing this to our attention. Upon reviewing Figure 1, we acknowledge that the legend labeled (a) and other textual indications appear blurred in comparison to Figure b. We sincerely apologize for any inconvenience this may have caused. To rectify this issue, we have revised Figure 1to include clearer and more readable indications. We deeply appreciate your feedback and will ensure that all figures in our subsequent submissions adhere to the necessary standards for clarity and readability.

Comment 5:  In general, the sentences (throughout the manuscript) seem longer, please pay attention to it (my personal suggestion).

Response:  Thank you for your feedback on the manuscript. We highly value your personal suggestion and have taken it into careful consideration. Upon revising the manuscript, we have concerted efforts to vary sentence length and structure in order to enhance readability and flow. If you have any specific suggestions or can point out examples where you think shorter sentences would be more effective, please do not hesitate to share them with us. Your input is invaluable and greatly appreciated as we strive to continually improve the overall quality of the manuscript.

Comment 6: A few sentences are still confusing, I think, minor English editing is recommended.

Response: Thank you for your feedback. We understand that some sentences may still be confusing. We appreciate your suggestion for minor English editing to clarify these points. We reviewed the text carefully again and made the necessary adjustments to ensure clarity and readability. If you have any specific suggestions or areas you'd like me to focus on, please let me know. Your input is invaluable in helping me improve the quality of the content.